# A Vaccine Displaying a Trimeric Influenza-A HA Stem Protein on Capsid-Like Particles Elicits Potent and Long-Lasting Protection in Mice

**DOI:** 10.3390/vaccines8030389

**Published:** 2020-07-15

**Authors:** Susan Thrane, Kara-Lee Aves, Ida E. M Uddbäck, Christoph M. Janitzek, Julianna Han, Yuhe R. Yang, Andrew B. Ward, Thor G. Theander, Morten A. Nielsen, Ali Salanti, Allan R. Thomsen, Jan P. Christensen, Adam F. Sander

**Affiliations:** 1Department of Immunology and Microbiology, University of Copenhagen, 2200 Copenhagen, Denmark; susanthrane@gmail.com (S.T.); kara-lee@sund.ku.dk (K.-L.A.); iuddback@sund.ku.dk (I.E.MU.); christoph@sund.ku.dk (C.M.J.); thor@sund.ku.dk (T.G.T.); mortenn@sund.ku.dk (M.A.N.); salanti@sund.ku.dk (A.S.); athomsen@sund.ku.dk (A.R.T.); jpc@sund.ku.dk (J.P.C.); 2Department of Integrative Structural and Computational Biology, The Scripps Research Institute, La Jolla, CA 92037, USA; juliannahan@scripps.edu (J.H.); yryang@scripps.edu (Y.R.Y.); andrew@scripps.edu (A.B.W.); 3AdaptVac Aps, Agern Alle 1, 2970 Hørsholm, Denmark

**Keywords:** universal influenza vaccine, HA-stem antigen, capsid-like particles, Virus-like particles, trimer display, IgG2a, pandemic preparedness

## Abstract

Due to constant antigenic drift and shift, current influenza-A vaccines need to be redesigned and administered annually. A universal flu vaccine (UFV) that provides long-lasting protection against both seasonal and emerging pandemic influenza strains is thus urgently needed. The hemagglutinin (HA) stem antigen is a promising target for such a vaccine as it contains neutralizing epitopes, known to induce cross-protective IgG responses against a wide variety of influenza subtypes. In this study, we describe the development of a UFV candidate consisting of a HA_stem_ trimer displayed on the surface of rigid capsid-like particles (CLP). Compared to soluble unconjugated HA_stem_ trimer, the CLP-HA_stem_ particles induced a more potent, long-lasting immune response and were able to protect mice against both homologous and heterologous H1N1 influenza challenge, even after a single dose.

## 1. Introduction

Influenza A is a respiratory virus causing up to 5 million cases of severe illness and between 290,000 and 650,000 influenza-related deaths every year [1]. Current influenza vaccines are directed against the major surface antigens, hemagglutinin (HA) and neuraminidase (NA). These vaccines generate strain-specific responses and therefore need to be re-designed annually in response to antigenic shift and drift [2,3]. This cumbersome process is further complicated by the current production methods, which are largely still reliant on traditional egg-based propagation. This process is slow, costly and severely restricts the number of doses that can be produced for a given flu season, meaning we are left largely unprepared for pandemic outbreaks [4,5,6,7]. Consequently, several alternative approaches are being pursued with the aim to develop a universal flu vaccine (UFV), ideally offering life-long protection against a broad range of viral strains and subtypes including emerging pandemic strains [8,9].

The HA protein is responsible for the recognition and entry of the virus into target cells and is integral to its infectivity. In the native confirmation, HA is a homotrimeric membrane glycoprotein composed of a hypervariable head and a more conserved stem region. Although the human immune system can react against both regions, most of the response is directed towards the variable head. In the context of developing a UFV, one promising strategy to increase the breadth of protection is to focus the immune response towards the more conserved HA-stem. This region is known to contain epitopes for broadly neutralizing antibodies (bnAb) and several HA-stem antigens have proven to elicit cross-protective antibody responses in preclinical animal models [10,11,12,13,14,15,16]. In addition to direct neutralization of the virus, antibodies directed against the stem region mediate a number of Fc-R-dependant functions including phagocytosis and antibody-dependent cellular cytotoxicity (ADCC), which contributes significantly to the overall protection [17,18,19]. Importantly, high concentrations of HA-stem-specific antibodies are required for adequate immunity. Therefore, the low intrinsic immunogenicity of the HA-stem region [20] and the fact that current HA-stem antigen designs contain immune-dominant, non-neutralizing epitopes (including neo-epitopes formed during their engineering) pose a significant challenge to the development of an HA-stem-based UFV. Therefore, optimal vaccine formulation of the HA-stem antigen is essential to maximize immunogenicity, engage both adaptive and innate immune functions, and favour display of bnAb epitopes over non-neutralizing epitopes.

On that basis, we investigated whether the immune response towards a HA-stem vaccine candidate that has previously shown to elicit bnAb responses in both mice and non-human primates [11], could be improved by presenting it on the surface of rigid capsid-like particles (CLP). Due to their size and repetitive surface geometry, which mimic that of live viruses, CLPs induce rapid lymph node drainage, enhanced innate immune activation and strong B-cell receptor cross-linking, leading to potent and long-lasting antibody responses [21,22]. In this study, CLP-display of HA_stem_ was achieved using the Tag/Catcher-AP205 vaccine platform [23,24,25,26]. Upon mixing of the antigen and CLP components, stable antigen:CLP complexes are formed by the spontaneous formation of isopeptide bonds between the reactive tag (SpyTag) and Catcher protein (SpyCatcher), which are genetically fused to the antigen and CLP subunit protein, respectively. Accordingly, the platform can facilitate unidirectional and high density display of even large and complex proteins [26] and has previously been demonstrated to increase immunogenicity and mediate a focused antibody response targeting specific parts of the displayed antigen [27].

Our results show that CLP-display of the trimeric HA_stem_ protein modified the isotype profile of the induced antibodies and significantly improved the protective capacity of vaccine-induced responses in mice challenged with both homologous and heterologous H1N1 influenza strains.

## 2. Materials and Methods

### 2.1. Design, Expression and Purification of HAstem Trimer Antigen

The HA_stem_ antigen was engineered by Impagliazzo et al. based on the H1N1 A/Brisbane/59/2007 sequence [11]. To enable CLP-display, the HA_stem_ sequence #4900 [11] was genetically fused at the C-terminus with a 6xhistidine tag followed by a Glycin-Glycin linker and the Spytag sequence (AHIVMVDAYKPTK) (Appendix A). The gene sequence was further modified to contain a BamHI restriction site at the N-terminus and a stop-codon followed by an EcoRI restriction site at the C-terminus. The codon optimized (Trichoplusia ni cells) gene sequence was finally synthesized by Geneart (Life Technologies).

To generate recombinant HA-stem influenza trimers, flashBAC™ Gold DNA (Oxford Expression Technologies, Oxford, UK) was co-transfected with pAcGP67A/HA_stem_ into Sf9 insect cells using Lipofectamine 2000 Reagent (Invitrogen, Carlsbad, CA, USA), according to manufactures instructions. Supernatant containing recombinant Baculovirus was harvested and used to generate a high-titer virus stock, for infection of High-Five insect cells. Infected High-Five cells were incubated for 16–28 h at 28 °C with shaking. The filtered supernatant was buffer exchanged (1 × PBS, pH 7.4, supplemented with 60 mM Imidazole) and concentrated using a QuixStand Benchtop system (10,000 MWCO hollow fiber cartridge, surface area 650 cm^2^ (GE Healthcare, Chicargo, IL, USA) followed by filtration through a 0.22 µm filter. The 6xhistidine tagged recombinant HA-stem protein was purified on a 5 mL HisTrap HP column (GE healthcare). Bound protein was eluted with 500 mM Imidazole in 1 × PBS buffer, pH 7.4. To isolate assembled HA_stem_ trimers, HA_stem_ protein fractions were further purified by size-exclusion chromatography using a Highload Superdex 200 pg chromatography column (GE healthcare). Fractions containing HA_stem_ trimers were identified by non-reducing SDS-PAGE. Protein concentrations were measured using a BCA kit (Thermo Fisher Scientific, Waltham, MA, USA).

### 2.2. Expression and Purification of CLP Platform, SpyCatcher-AP205

The expression and purification of SpyCatcher-CLP has previously been described [23]. Briefly, SpyCatcher was genetically fused to the N-terminus of the bacteriophage AP205 coat protein (Gene ID: 956335). The SpyCatcher-CLP gene sequence was cloned in a pET-15b vector, using NcoI and NotI restriction sites, and recombinant expression was performed in *E. coli* One Shot^®^ BL21 Star™ (DE3) cells (Invitrogen). Purification of SpyCathcer-CLP was performed by ultracentrifugation over an Optiprep™ (Sigma-Aldrich, St Lois, MO, USA) density step gradient.

### 2.3. Conjugation of HA_stem_ Trimer to CLP Platform

To develop CLP-HA_stem_, SpyTagged HA_stem_ trimers and SpyCatcher-CLP were mixed in a 1:1 molar ratio and incubated overnight at 4 °C. Unbound excess HA-stem antigen was removed by density step-gradient ultracentrifugation. The CLP-HA_stem_ was finally dialyzed against a 1xPBS buffer, pH 7.4, using a 1000 kDa MWCO dialysis tube (Spectrum Labs, San Francisco, CA, USA).

### 2.4. Quality Assessment of the HA_stem_ CLP Vaccine

The CLP-HA_stem_ vaccine was assessed via Dynamic Light Scattering (DLS) as described previously [28]. In brief, the vaccine was spun at 15,000 g for 10 min, before being loaded into a Eppendorf Uvette cuvette (Sigma-Aldrich, St Louis, MO, USA) and measured 20 times at 25 °C on a DynoPro NanoStar (WYATT Technology, Santa Barbara, CA, USA) using a 658 nm laser. Dynamics software, Version 7.5.0 (WYATT Technology, Sanata Barbara, CA, USA) was used to estimate the intensity-average diameter (nm) and percentage polydispersity (%Pd). CLP-HA_stem_ was further visualized by negative stain Electron Microscopy as described in the subsequent section.

### 2.5. Negative-Stain Electron Microscopy Sample Preparation, Data Collection, and Processing

HA_stem_ trimers were mixed with MEDI8852 Fab fragments at a molar ratio of 1:3 HA trimer to Fab fragment for 1 h at room temperature. Complexed HA_stem_:MEDI8852 samples and CLP-HA_stem_ samples were diluted to 0.01 mg/mL with 1× TBS, and then 3 μL of sample was immediately applied to glow-discharged 400 mesh carbon-coated Cu grids. Grids were subsequently stained with 2% (wt/vol) uranyl formate for 35–45 s and then blotted using filter paper until completely dry. All grids were imaged on a 120 kV FEI Tecnai Spirit electron microscope (FEI, Hillsboro, OR, USA) using a nominal magnification of ×52,000, resulting in 2.05 Å/px. Micrographs were collected using a TVIPS TemCam-F416 (4 k × 4 k) camera with the Leginon interface [29] and a defocus of 1.5 μm. Particles were selected using a Difference of Gaussians (DoG) picker in Appion [30], extracted, and classified into 2D class averages with Relion v2.1 [31].

### 2.6. Bio-Layer Interferometry

CR9114 and S9-3-37 IgG at 5 μg/mL or MEDI8852, C05, and HIV PG9 Fab fragments at 20 μg/mL were loaded onto anti-human IgG Fc Capture (AHC) or anti-human Fab CH1 (FAB2G) biosensors (ForteBio, Fremont, CA, USA) in kinetics buffer (0.01% BSA + 0.002% Tween20 in 1× phosphate-buffered saline) and dipped into wells containing HA_stem_ (135 nM or 270 nM), CLP-HA_stem_ (250 nM or 500 nM total protein), or kinetics buffer alone using an Octet Red96 instrument (ForteBio). After loading, association was measured for 700 s, followed by dissociation for 750 s in kinetics buffer. A baseline containing kinetics buffer was subtracted from each data set, and curves were aligned on the y axis using the baseline step.

### 2.7. Animal Immunization Studies

Female Balb/cA mice, 6–8 weeks old, were obtained from Janvier Labs and housed in a specific pathogen–free facility. Before being used in experiments, animals were allowed to acclimatized at the facility for at least 1 week.

Endotoxins were removed from CLP-HA_stem_ vaccine formulations as previously described [32], and formulated with AddaVax^TM^ (InvivoGen, San Diego, CA, USA) according to the manufacturer’s instructions. Mice were immunized intramuscularly with either CLP-HA_stem_ or with a vaccine formulation containing soluble HA_stem_ trimers. An antigen dose-escalation study was conducted by vaccinating mice with 1, 5 or 10 µg HA_stem_ (soluble or CLP displayed) following prime-boost-boost regimen with a 3-week interval. Influenza virus challenge was performed 1 month after the last boost using the heterologous strain A/Puerto Rico/8/1934 (H1N1/PR8). For the longevity-study, mice were vaccinated every 3rd week with 4.5 µg HA_stem_ (CLP display vs. soluble) in a prime-boost-boost setting and challenged 35 weeks after the first immunization with the homologous influenza strain H1N1A/Brisbane/59/2007 (H1N1/Brisbane). Finally, for the one-shot study, mice were vaccinated once with 4.5 µg HA_stem_ (CLP display vs. soluble) and 1 month after the final vaccination mice were challenge with PR8 or Brisbane strain.

### 2.8. Serum Immunoglobulin Levels

To assess the vaccine-induced antibody responses, enzyme-linked immunosorbent assay (ELISA) was used to measure HA_stem_-specific antibody levels, as previously described [23]. In brief, 96-well microtiter plates (Nunc MaxiSorp, Inivtrogen) were coated with 0.1 μg/well recombinant HA_stem_ protein, which was then incubated with a serial dilution of serum. Plates were probed with the relevant HRP-conjugated secondary antibody (i.e., anti-mouse IgG/IgG1/IgG2a/IgG3) and then developed with a tetramethylbenzidine (TMB) substrate and the absorbance subsequently measured at OD 450 nm. For the longevity study, the geometric means of the titers (GMT) were calculated on all bleeds. For all other studies, the endpoint titers are reported and unless otherwise stated in the figure, a cut off of OD 0.1 was applied.

### 2.9. Virus Challenge

The influenza viruses H1N1 Brisbane A/59/07 (Brisbane) and H1N1 A/Puerto Rico/8/34 (PR8) were used for sub-lethal challenge studies as described previously [33,34]. An amino acid sequence comparison of the HA_stem_ protein between the two viral strains can be seen in Appendix A. For each virus preparation, the lethal dose was determined, and 1–3 LD_50_ used. For Brisbane challenge, this corresponded to 100,000 PFU and for PR8 challenge 100 PFU was used. Mice were anaesthetized using an i.p. injection of 250 mg/kg avertin (2,2,2 tribromoethanol in 2-methyl-2-butanol), and then infected i.n. with 30 μL of appropriately diluted influenza virus. Following influenza challenge, mice were assessed by daily weights.

### 2.10. Influenza Virus Plaque Assay

Three or five days post influenza challenge, mice were euthanized and the lungs harvested and frozen via cryopreservation. Lungs were homogenized using sterilized sand and a mortar and pistil. PBS + 1% FBS was added to the samples to obtain a 10% weight/volume suspension and then spun for 15 min at 600 G and 4 °C. Supernatant was collected and kept on ice until use.

MDCK plaque assay was performed as previously described [33]. Briefly, 4.5 × 10^4^ MDCK cells were grown in 96-well plates overnight in 100 μL complete medium. Ten-fold dilutions of the lung suspensions were prepared using an influenza growth medium containing DMEM 1965 medium with 0.2% BSA, 2 mM L-glutamin, 200 IU/mL penicillin, 50 μg/mL streptomycin, 1% sodium-pyruvate, and 5 units/mL TPCK Trypsin. MDCK-cells were washed twice with PBS and then incubated with 50 μL of each virus dilution for 2 h at 37 °C, 5% CO_2_. Media was then removed, and the cells were overlayed in a medium containing 2× minimum essential medium (MEM) eagle supplemented with 0.4% BSA, 10% NaHCO3, 2% Streptomycin, 2% penicillin and 5 units/mL TPCK trypsin mixed 1:1 with 1.8% methylcellulose. Following a 48 h incubation at 37 °C and 5% CO_2_, the overlay was removed and the wells were washed twice with PBS. Cells were fixated with 4% formaldehyde for 30 min at room temperature and then washed twice with PBS and permeabilised with warm 0.5% Triton-X in Hanks balanced salt solution medium. Cells were subsequently washed and then incubated with primary α-influenza nucleocapsid A mAb (Nordic Biosite, Copenhagen, Denmark) diluted 1:1500 in PBS + 1% BSA for 1 h at 37 °C and 5% CO_2_. The antibody was removed and cells washed five times. This was followed by incubation with a goat α-mouse HRP conjugated secondary mAb (Dako, Glostrup, Denmark) at a 1:500 dilution in PBS + 1% BSA for 1 h at 37 °C, 5% CO_2_. Cells were subsequently washed five times with PBS before adding 200 μL substrate solution containing 3 mg/mL 3-amino-9-ethylcarbazole, 0.07% H_2_O_2_ and 5 mM citrate phosphate buffer (pH 5) to the wells for 30 min at room temperature. The substrate was removed and cells were washed before counting. All samples were run in duplicates. Plaque forming units per g lung tissue was calculated according to the following formula:Dilution factor × average number of plaques/well × 20 = PFU/gDetection level was calculated to be 500 PFU/g (2.7 log PFU/g)

### 2.11. Ethical Statement

All animal experiments were approved by the Danish Animal Experiments Inspectorate, approval numbers: 2018-15-0201-01541 and 2015-15-0201-00623, and were conducted in accordance with national Danish guidelines. Mice were housed in an AAALAC accredited facility in accordance with good animal practice as defined by FELASA.

### 2.12. Statistical Analysis

To determine statistical significance of ELISA endpoint titers and the plaque assays, pairwise comparisons of immunization groups was performed using an unadjusted, non-parametric, two-tailed, Mann–Whitney Rank Sum Test, with a statistical significance defined at *p* < 0.05.

## 3. Results

### 3.1. Development and Characterization of a CLP Vaccine Displaying a Trimeric HA_stem_ Protein

A previously described HA_stem_ protein, based on A/Brisbane/59/2007 (H1N1) influenza virus [11], was genetically modified to contain a C-terminal SpyTag and subsequently expressed and purified from baculovirus-transfected insect cells. The Spytagged HA_stem_ formed stable disulfide-linked trimers, similar to the original HA_stem_ protein (Figure 1a and Appendix A). To further characterize the protein, the presence and accessibility of known bnAb epitopes was confirmed by Bio-layer interferometry using MEDI8852 [35], CR9114 [36] and S9-3-37 [37] monoclonal antibodies (Figure 1b and Appendix A). Electron microscopy of HA_stem_- MEDI8852 complexes revealed that three Fab molecules bound to each trimer (Figure 1b). These analyses demonstrated that well-characterized bnAb epitopes were retained in the SpyTag-modified HA_stem_ recombinant protein, and that the protein shares a similar structure with the native HA surface antigen. Mixing of Spytagged HA_stem_ trimers with SpyCatcher-CLP resulted in the formation of HA_stem_ CLPs, each displaying approximately 20–30 HA_stem_ trimers per particle (Figure 1c,d). Excess unconjugated HA_stem_ antigen could efficiently be removed by ultracentrifugation (Figure 1d and Appendix A). Interestingly, the majority of HA_stem_ trimers were anchored to the CLP using multiple (i.e., two or three) of the Spytags present at the C-terminus of each protomer. This was evidenced by the small percentage of surface displayed HA_stem_ protomers that was not covalently bound to a CLP subunit (Figure 1d lane 3). Dynamic light-scattering (DLS) analysis and transmission electron microscopy revealed that HA_stem_ CLPs were largely monomodal and monodispersed (polydispersity of 14.3%) and had a larger diameter (58 nm) than the unconjugated SpyCatcher-CLP (46 nm) (Figure 1e and Appendix A). The CLP-HA_stem_ contained bacterial RNA, which is encapsulated during its assembly inside the bacterial production cell (Appendix A). All vaccine formulations had low endotoxin levels (<3 EU/mL). Bio-layer interferometry analysis of the CLP-HA_stem_ particles showed that high-density display of the trimers did not compromise the accessibility of known broadly neutralizing epitopes (Appendix A). Together, these results suggested that the Tag/Catcher AP205 system could be used to display the HA_stem_ antigen in a CLP structural format, allowing further comparative studies (CLP-HA_stem_ versus soluble HA_stem_) to assess the potential benefits of multivalent CLP antigen display.

### 3.2. Increased Immunogenicity of CLP-HA_stem_ Offers Potential Dose Sparring

The immunogenicity and protective capacity of the CLP-HA_stem_ particles was compared head-to-head with a similar vaccine formulation containing soluble HA_stem_ trimer. In a dose-escalation study, mice were immunized with a prime, boost, boost regime, and vaccine-induced anti-HA_stem_ antibody levels were measured 2 weeks after each vaccination (Figure 2a). After the first immunization (measured at week 2), all mice vaccinated with CLP-HA_stem_ had a high antibody response, which was not affected by the antigen dose. In comparison, mice receiving soluble HA_stem_ protein produced significantly lower antibody responses (*p* < 0.005) and 6 out of the 10 mice receiving the lowest dose failed to generate a detectable anti-HA_stem_ titer (Figure 2b). The second vaccination with the CLP-HA_stem_ vaccine (measured at week 5) led to a 2-log increase in ELISA antibody titers, again irrespective of the antigen dose, and this level of anti-HA_stem_ antibodies did not increase substantially after the third vaccination (measured at week 8). In comparison, only mice that received the highest dose of soluble HA_stem_ antigen produced a similarly high level of anti-HA_stem_ antibodies after the second vaccination, whereas antibody levels in mice receiving lower doses (1 µg and 5 µg) needed three immunizations to reach this titer (Figure 2b). Serum from naïve mice showed no reactivity against the HA_stem_ trimer (data not shown).

To assess the protective efficacy of the vaccines, mice were challenged with a heterologous H1N1 influenza virus, A/Puerto Rico/8/1934 (H1N1/PR8) 4 weeks after the final immunization. Five days later, mice were euthanised and the virus burden in the lungs was evaluated in a MDCK plaque assay. Although anti-HA_stem_ antibody levels were within a similar range between the different vaccination groups after three immunizations (Figure 2b), significant differences were seen in the virus burden following the challenge (Figure 2c). Mice vaccinated with the CLP-HA_stem_ vaccine had a significantly lower viral load compared to the naïve control mice (*p* < 0.005), irrespective of the antigen dose. By contrast, only animals vaccinated with the higher doses (5 and 10 µg) of soluble HA_stem_ trimer had reduced viral loads compared to the naïve controls. For both the CLP-HA_stem_ and soluble HA_stem_ vaccine, immunization with 5 or 10 µg antigen resulted in a significantly lower viral load when compared to the unvaccinated control group (Figure 2c).

### 3.3. CLP-HA_stem_ Elicits a Long-Lived Protective Response

An important quality of a future universal influenza vaccine is its ability to elicit a long-lasting protective response to eliminate the need for annual flu shots. With this in mind, the longevity of the response induced by the CLP-HA_stem_ vaccine was investigated. Mice were immunized with 4.5 µg antigen in a similar prime-boost-boost schedule as used in the dose-response study and thereafter, the mice were intermittently bled and HA_stem_ titer followed over a course of 35 weeks (Figure 3a).

No difference was observed in the geometric mean titer of anti-HA_stem_ IgG between mice vaccinated with soluble HA_stem_ trimer or CLP-HA_stem_ (Figure 3b). In both groups, the peak antibody titer was observed after the third immunization (measured at week 8). Hereafter, anti-HA_stem_ titers slowly declined in both vaccination groups between week 9 and week 20, where after the antibody levels remained at a stable plateau throughout the remainder of the study (measured up to week 29). The antigen-specific IgG isotype profile, measured 29 weeks after vaccination, revealed that while both groups had similar amounts of antigen-specific IgG1, mice immunized with CLP-HA_stem_ had a significantly higher amount of antigen-specific IgG2a (*p* < 0.005) (Figure 3c).

To assess the protective capacity of the vaccine-induced immune response, mice were challenged with a homologous influenza strain, H1N1 A/Brisbane/59/2007, 34 weeks after the first vaccination. Five days after influenza challenge, the lungs were harvested and viral load was measured in a plaque assay. This experiment showed that whereas both vaccines could reduce the viral load compared to the unvaccinated control mice, CLP-HA_stem_ vaccinated mice were significantly better protected against infection (median titer of 7500 PFU/g) compared to mice vaccinated with the soluble HA_stem_ vaccine (median titer of 500,000 PFU/g); *p* < 0.005) (Figure 3d).

### 3.4. A Single Vaccination with CLP-HA_stem_ Protects against Homologous and Heterologous Challenge

To assess the potency of the CLP-HA_stem_ vaccine, the protection induced after a single immunization was investigated (Figure 4a). Mice were vaccinated once with either soluble HA_stem_ or the CLP-HA_stem_ at a dose of 4.5 µg HA_stem_ protein. Three weeks after vaccination, mice receiving the CLP-HA_stem_ vaccine had significantly higher anti-HA_stem_ IgG titers compared to mice receiving the soluble antigen (*p* < 0.0005) (Figure 4b). Moreover, as in the previous experiment, mice immunized with the CLP-HA_stem_ vaccine had significantly higher levels of anti-HA_stem_ IgG2a antibodies (*p* < 0.0005), while the IgG1 antibody titer was comparable to mice vaccinated with soluble HA_stem_ (Figure 4c).

Five weeks post-immunization, mice were challenged with either the homologous (A/Brisbane) or heterologous (A/PR8) influenza virus, and lung viral load was determined in a MDCK plaque assay 3 and 5 days post-inoculation. Mice vaccinated with a single dose of CLP-HA_stem_ were protected against both viral strains (Figure 4d–g). These mice had a lower viral load in the lungs compared to unvaccinated mice (median titer 5 days post challenge of 3 × 10^3^ versus 1 × 10^6^ PFU/g (*p* < 0.005) and 2.5 × 10^4^ versus 1.9 × 10^7^ PFU/g (*p* < 0.005)) for homologous and heterologous challenge respectively. Moreover, the mean viral load in the lungs did not increase between days 3 and 5, indicating that CLP-HA_stem_ vaccination did not simply delay viral replication but rather protected against it (Figure 4d,e). Finally, CLP-HA_stem_ vaccinated mice showed no weight loss as a result of the either influenza challenge (Figure 4f,g). Mice vaccinated with the soluble HA_stem_ fared less well. Five days after homologous challenge, the mean lung viral burden was lower on day 5 compared to the control animals (*p* < 0.05), but there was no sign of protection against heterologous challenge, where virus burden and weight loss were similar to that observed in the unvaccinated control mice (Figure 4d–g).

These results showed that a single dose of the CLP-HA_stem_ vaccine provided protection against both homologous and heterologous influenza virus challenge in mice and that this vaccine outperformed a vaccine based on the soluble HA_stem_ trimer.

## 4. Discussion

An effective UFV offering protection against both seasonal and pandemic viral strains would greatly improve our position in the ongoing battle against influenza. While several antigen candidates with broadly neutralizing potential (e.g., HA-stem) have been identified [10,11,12,13,14,38,39], several hurdles to UFV development still need to be overcome. Firstly, while high anti-HA antibody levels are important for protection [20], it has so far been challenging to induce strong humoral responses against the conserved stem region. Additionally, the people in society that are most at risk of severe disease and death, such as the elderly, often have reduced immune responsiveness making successful vaccination a challenge [40]. Importantly, during pandemic situations, additional requirements such as rapidly induced protection (ideally after a single vaccination) and the possibility for dose sparing are also vital [41]. On this basis, an effective vaccine formulation and delivery platform that can facilitate a fast, strong and long-lasting immune response is essential for an effective UFV.

It is well documented that presentation of foreign antigens on the surface of a CLP in a repetitive and ordered format, mimicking that of a native pathogen, can greatly enhance the immune response elicited against the antigen [26,42]. In this study, we set out to test whether CLP presentation could increase the immunogenicity and protective efficacy of a promising HA_stem_ vaccine candidate. As a starting point, we used a HA_stem_ recombinant protein previously engineered to form stable trimers, exhibiting structural properties comparable to that of the full-length HA, and which has been validated to induce broad protection, even against heterosubtypic influenza strains [11].

For many viruses, such as HIV, coronaviruses and influenza, the immunologically important antigens exist as trimers. Therefore, in the context of developing effective vaccines, it is often essential to deliver the vaccine antigen in a quaternary structure similar to that of the native protein to ensure that important inter-subunit epitopes are displayed. However, display of multimeric antigens on CLPs through conventional means has long been technically challenging. Recently however, the Tag/Catcher AP205 platform was shown to facilitate the display of the large trimeric HIV envelope antigen [27]. In accordance, the present study has demonstrated the ability of the Tag/Catcher AP205 platform to facilitate high density, repetitive presentation of the HA_stem_ trimer in a unidirectional manner, while maintaining the accessibility of known bnAb epitopes. Additionally, the observed anchoring of multiple protomers in each HA_stem_ trimer to the CLP surface may help stabilize the native structure, thus potentially negating the need for complex engineering of the antigen, which can lead to the formation of unwanted neoepitopes.

Following the successful development of the CLP-HA_stem_ vaccine, we set out to assess whether CLP-display could improve the HA_stem_ antigen’s capacity for inducing a protective immune response in mice. This was done by running multiple immunization/challenge studies comparing head-to-head a CLP-HA_stem_ vaccine with a similar vaccine containing the soluble HA_stem_ protein. Collectively, the results indicate that CLP display leads to increased immunogenicity of the HA_stem_ antigen and mediate a faster induction of protective immune responses. The results also suggest that CLP-display even at low doses (i.e., 1 ug HA_stem_ antigen) elicits immune responses that lower the lung burden of virus after heterologous influenza challenge. It is also noteworthy that a single dose of CLP-HA_stem_ vaccine was sufficient to protect mice against heterologous influenza challenge. Although sterilizing immunity was not achieved, the vaccination resulted in a significantly lower viral load compared to vaccination with the soluble HA_stem_. The results are encouraging in the context of any viral pandemic situation, where rapidly induced protection from severe disease is required without the necessity for subsequent booster vaccinations. Equally important, the increased immunogenicity provides the prospect of dose sparing. The longevity study, which showed that elderly mice that had received the CLP-HA_stem_ were significantly better protected than mice receiving the benchmarking soluble vaccine, which further demonstrates the capability of the vaccine to induce long-lived protective responses.

Previous studies assessing soluble HA_stem_ vaccines have reported that while their stem-directed antibodies provide some protection against morbidity and mortality, the mode of action does not seem to be related to inhibition of viral replication or decreased viral loads [10,11]. In our study, however, immunization with CLP-HA_stem_ led to a significant reduction in viral titer compared to naive controls. This suggests that CLP display is modulating the immune response against the HA_stem_ in a different direction compared to conventional soluble vaccination.

Although the soluble and CLP-based vaccines induce similar overall antigen-specific antibody titers after a prime-boost-boost regimen, vaccination with CLP-HA_stem_ resulted in significantly lower viral loads compared to the soluble HA_stem_, following influenza challenge. This indicates that the quality of the humoral response and not the magnitude of the response was critical to the induced protection. This phenomenon has previously been observed [43,44,45] and could be due to several factors including (i) antibody specificity; (ii) antibody avidity; (iii) IgG subclass switching; and (iv) activation and synergy with other arms of the immune response including innate immunity and T cell responses.

An increasing body of evidence suggests that the in vivo protection induced by HA-stem-directed antibodies is largely mediated by antibody-dependent cellular cytotoxicity (ADCC) and is highly reliant on Fc-FcγR interactions [12,17,18,46]. Therefore, due to their relative affinities for activating versus inhibitory FcγRs and thus their different effector capabilities, IgG subclass plays a crucial role in influenza vaccine efficacy. In this study, while soluble vs. CLP display of the trimer did not result in a different total IgG titer after three immunizations, there was a clear difference in the IgG isotype profiles between the two groups, with the CLP-HA_stem_ vaccine inducing a significantly higher IgG2a titre. This isotype switching is likely due to the presence of prokaryotic RNA in the center of the CLPs, which is passively encapsulated during CLP expression. This RNA acts as a potent TLR7/TLR8 agonist and has been directly linked to increased IgG2a levels in other CLP-based vaccines [19,44,47]. In previous direct head-to-head studies, IgG2a monoclonal antibodies targeting the HA-stem [46] and M2e antigen [48] have been shown to induce a much more potent protection against lethal influenza challenge compared to IgG1 antibodies with the same epitope specificity. Thus, literature provides strong support for the role of IgG2a titer in CLP-HA_stem_-induced protection. However, more studies, possibly involving passive antibody transfer or the use of FcγR deficient mice, will be necessary to fully explore this hypothesis.

In combination with isotype switching, epitope specificity could also be contributing to the increased efficacy of CLP-HA_stem_. The unidirectional display facilitated by the tag/catcher conjugation strategy has previously shown to enable a more focused humoral response by masking certain regions of the antigen and promoting the accessibility or dominance of others [27]. While not directly evaluated in this study, the antibody repertoire induced by CLP-HA_stem_ may be focused on the more functionally relevant and vulnerable regions of the stem trimer, resulting in increased protection from viral challenge. Conversely, HA_stem_ in its soluble formulation may have induced antibodies predominantly targeting non-protective sites of the antigen, including the neoepitopes engineered into the trimer during its design. Detailed epitope mapping [49] of the antibody pools induced by each vaccine could reveal if this is indeed the case.

## 5. Conclusions

CLP display of the HA_stem_ trimer resulted in a potent, rapidly induced and long-lasting protective response against influenza challenge in mice; properties that would be of essential importance in a pandemic situation. Although further studies will be needed to fully reveal the mechanism underlying this enhanced efficacy, our results show that CLP-HA_stem_ is a promising universal flu vaccine candidate and supports the use of the Tag/Catcher AP205 platform in the development of vaccines targeting other trimeric viral antigens.

## Figures and Tables

**Figure 1 vaccines-08-00389-f001:**
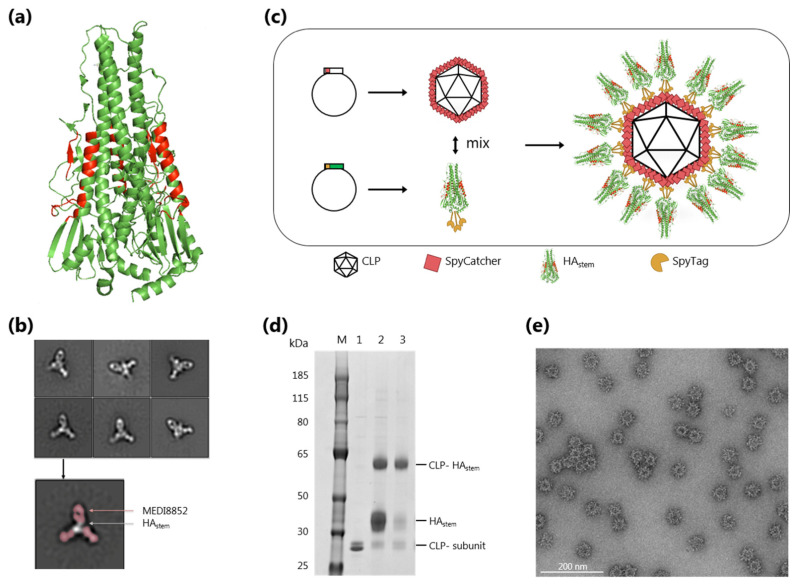
Design and characterization of CLP-HA_stem_ vaccine displaying known broadly neutralizing epitopes. (**a**) Structure of the HA_stem_ trimer, based on HA from A/California/04/2009 H1N1 (PDB ID 4M4Y). Amino acids corresponding to the head region of HA have been removed and epitopes of the bnAb’s MEDI8852, CR9114 and S9-3-37 are coloured red. (**b**) Representative 2D class averages of HA_stem_ trimer in complex with MEDI8852 Fab imaged using negative-stain EM. For one such image, 2D class densities attributed to the Fab are coloured red. (**c**) CLP-HA_stem_ vaccine development process. The stem region of a monomeric H1 hemagglutinin (HA) (genetically fused to Spytag at the C-terminus) was recombinantly expressed. The SpyCatcher-CLP and HA_stem_ trimer were expressed and purified separately and subsequently mixed. The tag/catcher covalent conjugation system ensures a vaccine with HA_stem_ trimers presented unidirectionally and at high density on the CLP. (**d**) Reduced SDS-PAGE analysis showing the coupling of the CLP (*lane 1*) to the HA_stem_ antigen, resulting in a band shift comprising of one CLP capsid unit covalently bound to one HA_stem_ (*lane 2*). After conjugation, excess uncoupled antigen was removed by density ultracentrifugation (*lane 3*). M = molecular weight marker. (**e**) Representative negative stain electron microscopy of CLP-HA_stem_ vaccine, showing uniform, non-aggregated particles of approximately 58 nm in diameter. Scale bar represents 200 nm length.

**Figure 2 vaccines-08-00389-f002:**
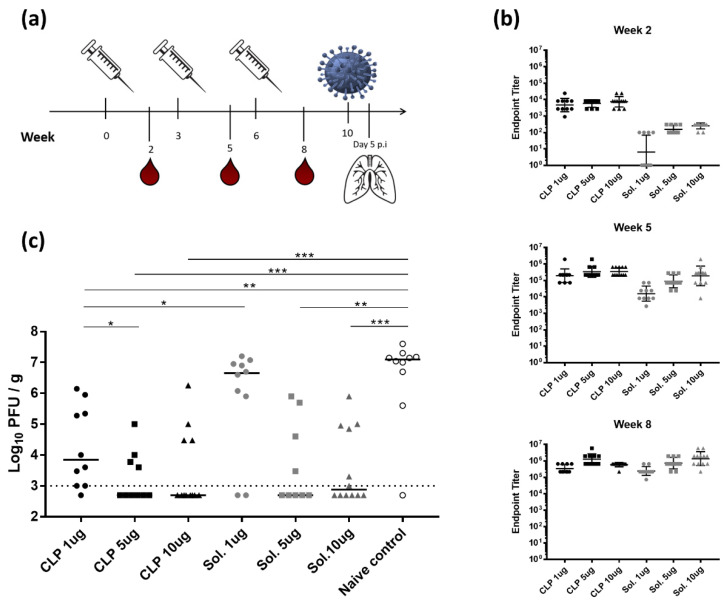
Dose-response study showing increased immunogenicity of CLP-HA_stem_. (**a**) Experimental setup. Mice (n = 10/group) were immunized three times with either CLP-HA_stem_ (CLP) or soluble HA_stem_ (Sol.) at a dose of 1, 5 or 10 µg at 3 week intervals. On weeks 2, 5 and 8, mice were bled. Four weeks after the final immunization, mice were challenged intranasally with H1N1 A/Puerto Rico/8/34 (A/PR8) and 5 days post inoculation (p.i) of influenza, mice were euthanized and their lungs harvested. (**b**) ELISA measurements of HA_stem_-specific IgG titres from serum taken at specified weeks. Cut off was set to OD_450nm_ of 0.2. Each dot represents one animal. Horizontal lines indicate geometric mean of the group and vertical lines indicate the standard deviation. Significance indicators are shown in Appendix A. (**c**) Lung viral titer measured using a MDCK plaque assay. Each dot represents one animal and horizontal lines represent the median titer per group. Dotted line represents the detection limit of the assay. * *p* < 0.05; ** *p* < 0.005; *** *p* < 0.0005.

**Figure 3 vaccines-08-00389-f003:**
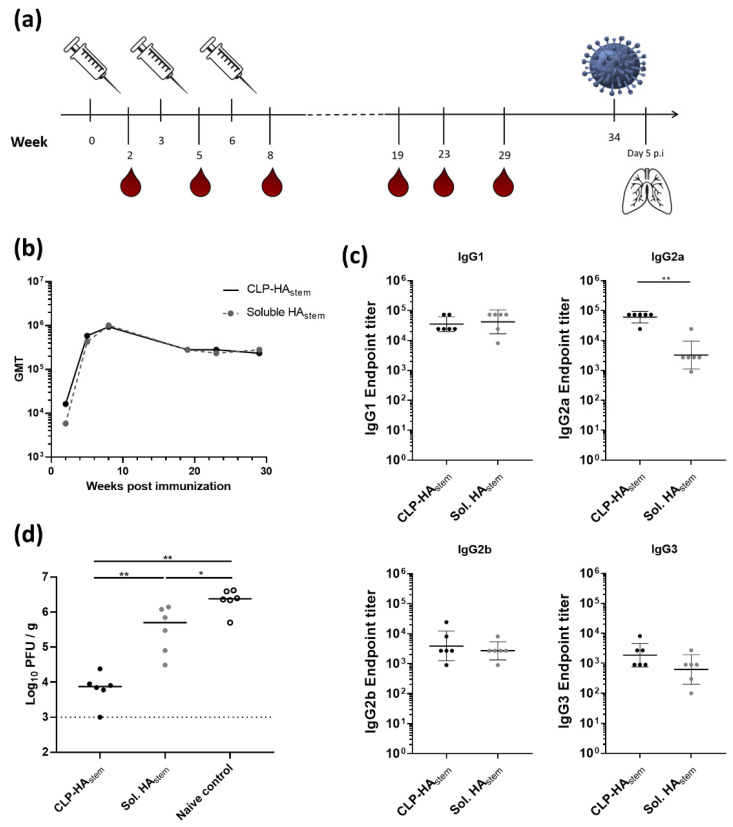
Long-lived protective response induced by CLP-HA_stem_. (**a**) Experimental setup. Mice (n = 6/group) were immunized three times with CLP-HA_stem_ or soluble HA_stem_ at 3 week intervals. Thiry-four weeks post prime vaccination mice were challenged intranasally with H1N1 Brisbane A/59/07 (A/Brisbane). Five days post inoculation (p.i) of virus, mice were euthanized and their lungs harvested. (**b**) ELISA measurements showing the geometric mean titer (GMT) of total IgG raised against HA_stem_ plotted against time post immunization. (**c**) ELISA measurements of individual anti-HA_stem_ IgG sub-class titres from serum taken at week 29. Each dot represents one animal. Horizontal lines indicate the geometric mean of the group and vertical lines indicate the standard deviation. (**d**) Lung viral titer was measured using a MDCK plaque assay. Each dot represents one animal. Horizontal lines represent the median titer per group. Dotted line represents the detection limit of the assay. * *p* < 0.05; ** *p* < 0.005.

**Figure 4 vaccines-08-00389-f004:**
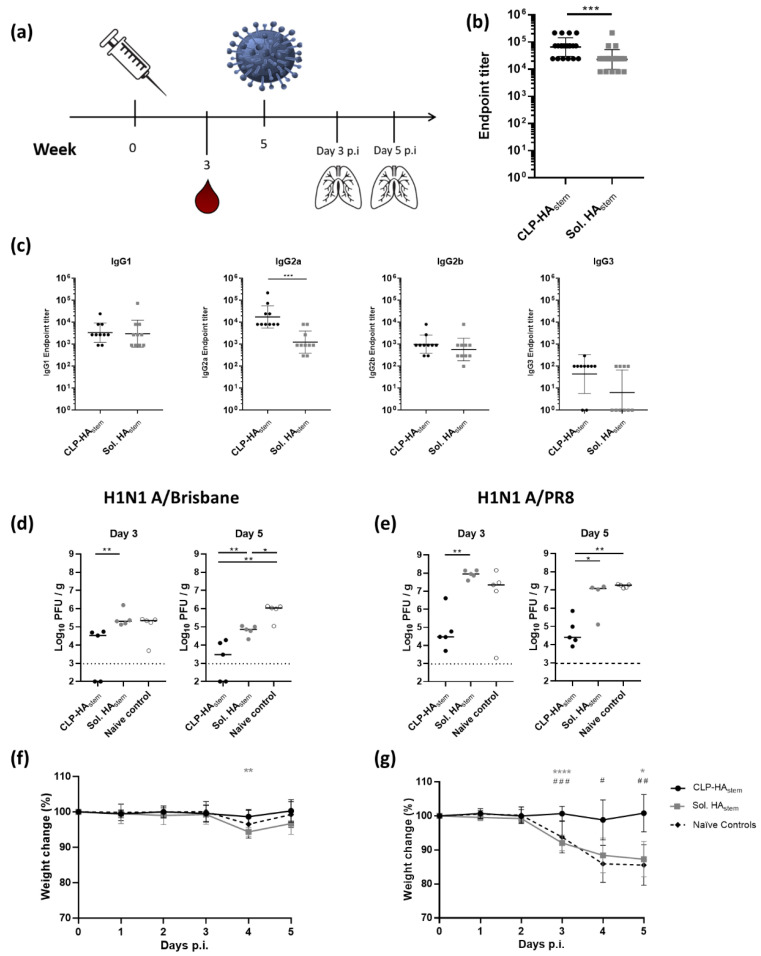
Single immunization protects against homologous and heterologous influenza challenge. (**a**) Experimental setup. Mice (n = 10/group) were immunized with CLP-HA_stem_ or Soluble HA_stem_ once. Blood was drawn at week 3. Five weeks post vaccination, mice were challenged intranasally with homologous (A/Brisbane) or heterologous (A/PR8) viral strains. Lungs were harvested on day 3 (n = 5/group) or 5 (n = 5/group) post influenza inoculation (p.i). (**b**) Anti-HA_stem_ endpoint IgG titer was measured by ELISA 3 weeks post immunization. (**c**) ELISA measurements of individual anti-HA_stem_ IgG isotype titers from serum taken at week 3. Each dot represents one animal. Horizontal lines indicate geometric mean of the group and vertical lines indicate the standard deviation (n = 10). (**d**) and (**e**) Lungs were isolated 3 or 5 days post challenge with A/Brisbane or A/PR8 respectively and viral titer was measured using MDCK plaque assay. Each dot represents one animal. Horizontal lines represent the median titer per group and the dotted line represents the detection limit of the assay. (**f**,**g**) Percentage weight loss following A/Brisbane and A/PR8 challenge respectively. Dots represent mean, and error bars represent standard deviation. * represents significant differences between the CLP-HA_stem_ group and the soluble HA_stem_ group. ^#^ represents significant differences between the CLP-HA_stem_ group and naïve controls. * *p* < 0.05; ** *p* < 0.005; *** *p* < 0.0005.

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
