# Peer review of "A Vaccine Displaying a Trimeric Influenza-A HA Stem Protein on Capsid-Like Particles Elicits Potent and Long-Lasting Protection in Mice"

_vaccines, 2020, doi:10.3390/vaccines8030389_

Round 1
Reviewer 1 Report
In the manuscript entitled: “A vaccine displaying a trimeric influenza-A HA stem protein on capsid-like particles elicits potent and long-lasting protection in mice”, the authors examined their experimental universal flu vaccine in mice. The topic is of obvious importance. The manuscript is easy to read. The presented data are clean and convincing. While the findings may appear to be incremental, it is a known challenge to improve immune responses towards a conserved stem region. Therefore, the authors successful attempt at increasing the immunogenicity of a HA-stem vaccine candidate is significant. However, I do have some concerns.
Major comments:
Survival is a gold standard for assessing vaccine effectiveness in preclinical studies and yet no survival data is presented in this report. The authors stated in the method&materials section that 1-3 LD50 was used for challenge experiments. However, it is puzzling that only weight loss for upto 5 days post infection are shown. Furthermore, based on Figure 4F, the challenge dose must have been lower than 1-3 LD50 since the maximum weight loss for naïve control (unvaccinated) is at most 5% on day 4. I would recommend doing a survival experiment using at least a LD50 dose to show that vaccination with CLP-HA stem provides superior protection.
Minor comments.
Since some epitopes in the HA are conserved among different subtypes, I would encourage authors to test their vaccination approach against heterosubtypic challenges. If authors could demonstrate that their vaccination approach is effective against heterosubtypic challenges, this would increase the significance of this study.
On day 5 post infection, no significance difference in weight loss was observed among different groups (Fig 4F). Does this mean that the differences observed in viral burden on day 3 and day 5 are biologically insignificant? Were the doses used for viral burden experiments and morbidity experiments equivalent?
The PR8 viral loads in Fig 4E are comparable between Sol. HA stem and naïve control groups and yet high Ig endpoint titers were detected in Sol. HA stem group. If Ig endpoint titer is a good correlate of cross-protection, one would expect to see a significant reduction in viral burden in Sol. HA stem group. I would recommend hemagglutination inhibition and/or viral neutralization assays if samples are still available.
The authors used uncongjuated HAstem trimer as a control but what would have been informative is to use commercially available vaccines for a comparison in a side-by-side experiment to show that the experimental vaccine is superior to what’s currently available.
The authors state that “high anti-HA antibody levels are required for protection”-line 355. While it is true that overwhelming evidence suggest that anti-HA antibody contributes to the overall protection against influenza, it is still not certain whether anti-HA is a good correlate of cross-protection for universal flu vaccines. In fact, significant effort has also been dedicated to develop universal flu vaccines based on T cell responses. Therefore, the authors’ statement seems a bit too strongly worded. I would recommend re-wording.
Since the entry point for influenzas virus is the mucosal surfaces of the lungs and upper airways, it would have been nice to see antibody levels, especially IgA, in the bronchoalvelar lavage fluid.
In order to fully appreciate the significance of high antibody endpoint titers in vaccinated groups, the naive serum should also be tested to show what the baseline level is ( at least in one graph).
Reviewer 2 Report
Major Comment:
A comparison of the amino acid sequence of HA2 domains of A/Brisbane and A/PR/8 should be included in the manuscript.
Figure 4C: What HA were the serum samples tested against and how were they tested? ELISA? They should be tested against both Brisbane and PR8 HA.
Minor Comments:
General: Please list the timepoints when samples were collected in the text. I.e. line 265 “The second vaccination” should list the days or weeks post prime.
Figure 2b: Antibody levels appear even at 8 weeks. Significantly different at this time?
Lines 352-353: Please avoid using hyperbole.
Line 355: The Authors say “central” do they mean “several”.
Line 419: The Authors state the CLP-HAstem “outperformed” the soluble antigen in terms of “protective efficacy”. This statement is nebulous and needs to be changed to reflect the clinical data presented in the paper, a reduction in weight-loss following heterologous challenge.
Lines 438-441: This sentence is tricky to read. Please split up into two sentences of shorten for clarity.
Round 2
Reviewer 1 Report
Comments:
If lethal challenge is not allowed, how did the authors determine the lethal dose 50 (line 171: “the lethal dose was first determined”? The median lethal dose (LD50) is defined as the dose that kills half of the tested population.
In addition to stating “1-3 LD50” – line 171, please include the actual PFU doses so that readers can better appreciate the robustness of the observed vaccine mediated protection. Also, this way the readers can judge for themselves whether the challenged dose is higher or lower than “what is expected to be the case in the context of human infection”.
If mice had to be euthanized prior to “fatal disease”, this should be stated in the method section and also define “fatal disease”. Please describe how mice were objectively decided to be euthanized. Was there a clearly defined end point (most commonly used end point is ~25% weight loss) that is objective and relevant? Please expand the “2.9 Virus challenge” to include all the details regarding how survival experiments were performed.
Author Response
If lethal challenge is not allowed, how did the authors determine the lethal dose 50 (line 171: “the lethal dose was first determined”? The median lethal dose (LD50) is defined as the dose that kills half of the tested population.
We know the LD50 due to our historical data from before lethal challenge was banned and find that it is the most relevant information regarding infection of animals, but we have now also included the pfu used for infection (see comment below).
In addition to stating “1-3 LD50” – line 171, please include the actual PFU doses so that readers can better appreciate the robustness of the observed vaccine mediated protection. Also, this way the readers can judge for themselves whether the challenged dose is higher or lower than “what is expected to be the case in the context of human infection”.
We have now included the PFU doses. This section now reads-
Line 172-174 “ For each virus preparation, the lethal dose was determined, and 1-3 LD50 used. For Brisbane challenge, this corresponded to 100,000 PFU and for PR8 challenge 100 PFU was used.”
If mice had to be euthanized prior to “fatal disease”, this should be stated in the method section and also define “fatal disease”. Please describe how mice were objectively decided to be euthanized. Was there a clearly defined end point (most commonly used end point is ~25% weight loss) that is objective and relevant? Please expand the “2.9 Virus challenge” to include all the details regarding how survival experiments were performed.
In line 171 we have made it clear that “sub-lethal challenge” was performed.
All our animal experiments are designed with the humane endpoint that animals cannot lose more than 25% of initial weight. However, this is not an issue in this study since we are not making lethal challenge as we end the experiment so early that animals are not dying, and hence we have not included this in the M&M section. In this study all animals were assign a day of termination (ie day 3 or 5- which is clearly stated both in the methods section as well as the results section and all the relevant figures), and were culled on the indicated days.
Please see attached
